# A New Strain of *Lecanicillium uredinophilum* Isolated from Tibetan Plateau and Its Insecticidal Activity

**DOI:** 10.3390/microorganisms10091832

**Published:** 2022-09-14

**Authors:** Yun Meng, P. I. Damithri Wellabada Hewage Don, Dun Wang

**Affiliations:** Institute of Entomology, Northwest A&F University, Xianyang 712100, China

**Keywords:** *Lecanicillium uredinophilum*, growth rate, condition of culture, insecticidal activity

## Abstract

A new strain QHLA of *Lecanicillium uredinophilum* was isolated from a Chinese caterpillar fungus complex and its optimum growth temperature and fermentation conditions were studied. Its insecticidal activity was tested against larvae of seven different insect pests, including *Henosepilachna vigintioctopunctata*, *Spodoptera exigua*, *Plutella xylostella*, *Spodoptera frugiperda*, *Sitobion avenae*, *Hyalopterus perikonus*, and *Aphis citricola*. The optimum growth temperature was 21–24 °C. The highest spore production of the strain QHLA was 4.08 × 10^6^ spore/mL on solid medium with a nitrogen source of NH_4_Cl. However, the highest mycelial growth rate of the strain QHLA was on solid medium with a nitrogen source from yeast extract (4.63 ± 0.03 mm/d). When the liquid medium contained peptone, yeast extract, and glucose, the water content of the mycelia was the lowest, while the spore production was the highest until day 12. When the liquid medium contained glucose, tussah pupa powder, KH_2_PO_4_, and MgSO_4_, the mycelia production was highest until day 8. The bioassay for insecticidal activity indicated that the LC_50_ values of QHLA were 6.32 × 10^3^ spore/mL and 6.35 × 10^3^ spore/mL against *Sitobion avenae* and *Aphis citricola*, respectively, while the LC_90_ values of the strain QHLA against *Aphis citricola* and *Sitobion avenae* were 2.11 × 10^7^ spore/mL and 2.36 × 10^8^ spore/mL, respectively. Our results demonstrated that the strain QHLA was a high virulence pathogenic fungus against insect pests, with the potential to be developed as a microbial pesticide.

## 1. Introduction

The *Lecanicillium* sp. is an important entomopathogenic fungus, and it is widely used in biological control in the field and plays a major role in the control of pests and nematodes [1]. It has been derived from number of sources, including soil, plants, decaying wood, and muscardine cadavers [2]. Previous studies have reported the separation and identification of *Lecanicillium* sp., including *L. muscarium*, *L. psalliotae*, *L. lecanii*, *L. gracile*, *L. coprophilum*, *L. rubum*, and *L. testudineum* [3,4,5,6].

Some of the fungi in the genus *Lecanicillium* are known to be effective against aphids, for example, *L. longisporum* is used for the control of *Myzus persicae* [7]. The secondary metabolites of *L. attenuatum* were effective against *Aedes albopictus* and *Plutella xylostella*, and its culture soup has higher insecticidal activity than mycelia cake [8]. *L. antillanum* that was isolated from soil had a negative effect on the egg structure of the nematode *Meloidogyne incognita* by secreting chitinase [9]. *L. lecanii* was isolated from within asymptomatic leaves of cotton, and insect bioassays demonstrated that it significantly reduced the rate and period of reproduction of *A. gosspy* [10]. *L. lecanii* also has the potential to control *B. tabaci* significantly [11], and the partially purified protein of the *L. lecanii* enhances the plant resistance to *B. tabaci* by regulating the content of secondary metabolites in the jasmonic acid (JA) and salicylic acid (SA) metabolic pathways of cotton plants [12]. *L. muscarium* is known to control the adult and larval stages of houseflies [13], *Aleurodicus rugioperculatus*, *Encarsia guadeloupae* [14], *B. tabaci*, *Trialeurodes vaporariorum* [6], and the sweetpotato whitefly [15]. The *L. uredinophilum* of *Lecanicillium* sp. was isolated from rusts and chitinous bodies, but its culture condition and insecticidal activity have not been reported [16,17].

The main objective of this study was to identify the strain QHLA isolated from the fresh samples of Chinese caterpillar fungus complex and to clarify the biological characteristics of the strain QHLA, especially for its optimal growth temperature and fermentation conditions. In addition, the potential pesticide application of QHLA was evaluated by using *Henosepilachna*
*vigintioctopunctata* (Coleoptera Coccinellidae), *Spodoptera exigua* (Lepidoptera Noctuidae), *Plutella xylostella* (Lepidoptera Plutellidae), *Spodoptera frugiperda* (Lepidoptera Noctuidae), *Sitobion avenae* (Hemiptera Aphididae), *Hayloperus perikonus* (Hemiptera Aphididae), and *Aphis citricola* (Hemiptera Aphididae) [18,19,20,21,22]. This research will provide a better understanding of the optimal growth temperature, carbon and nitrogen sources, the composition of the fermentation broth, and the effective pesticidal range of the QHLA and provide a theoretical basis for its production and application in the field.

## 2. Materials and Methods

### 2.1. Specimens Collection and Isolation

Samples were collected from the Tibetan Autonomous Prefecture of Guoluo, Qinghai Province in May 2019. The specimens were isolated by 1/4 SDAY medium containing antibiotics (peptone 2.5 g/L, D-glucose anhydrous 10 g/L, yeast extract 2.5 g/L, streptomycin 0.6 g/L, kanamycin 0.6 g/L, agar 15 g, 1000 mL of water), then cultured in an artificial incubator at 15 °C for 15 d. The edges of the colonies were selected and purified in the new 1/4 SDAY medium.

#### 2.1.1. Morphological Observation of Strain

The purified strain was cultured in 1/4 SDAY medium at 15 °C and shaded for 30 d. The growth characteristics of the strain were observed, including colony morphology, mycelial color, exudate color, and growth rate. The morphological characteristics of strain were observed by microscope (S/N:C 1709110263, SOP TOP; Ningbo Sunny Instruments Co., Ltd., Zhejiang, China), mainly including mycelial characteristics, sporulation structure characteristics, spore morphology, size, and the isolated strain was identified.

#### 2.1.2. Extraction and Amplification of Genomic DNA

The mycelia for DNA extraction were harvested from plated colonies using a sterile needle and placed in a 1.5 mL centrifuge tube. The genomic DNA was isolated using the method described by Aljanabi [23].

The purified DNA was used as a template to amplify the target genes using polymerase chain reaction (PCR). The PCR amplification was performed in 50 μL volumes containing 25 μL PCR mixture (GenStar, Beijing, China), 21 μL of twice-sterilized water, 1 μL of each primer, and 2 μL DNA template. The internal transcribed spacer (*ITS*), large subunit ribosomal RNA (nr*LSU*), small subunit ribosomal RNA (nr*SSU*), RNA polymerase II largest subunit (*RPB1*), and RNA polymerase II second largest subunit (*RPB2*) [24,25,26,27,28] were amplified with the primers and procedures mentioned in Appendix A. The PCR products were sequenced by Sangon Biotech (Shanghai, China).

#### 2.1.3. Phylogenetic Analysis

The sequences were submitted to GenBank (accession numbers were listed in Appendix A) and compared with available data from the GenBank database (closest identified relatives based on a BLAST search and accessed on 23 June 2022, National Centre for Biotechnology Information website; https://www.ncbi.nlm.nih.gov/) [29]. The closely related and the other entomogenous fungus sequences obtained from GenBank were used for phylogenetic analysis to identify the fungi. Each sequence was an independent file, and MEGA X was used to align each sequence. Then, a cladogram was used for maximum likelihood (ML) analyses in PhyloSuite v1.2.1 [30,31,32], employing a BIC model for all partitions, and nodal support was assessed with nonparametric bootstrap using 1000 pseudo-replicates. *ITS*, *RPB1*, and *RPB2* were used for Pairwise sequence (p-)distance analysis. (P-)distance analysis was performed with MEGA X using the compute pairwise distances model inferred from distance test and variance estimation method with the bootstrap method. 

### 2.2. Biological Characteristics of the QHLA

#### 2.2.1. Determination of Optimized Culture Temperature for the Fungus

The isolated strain was cultured until the colony was a regular circle with a diameter of about 60.0 mm. Then, the 7.0 mm punch was used to drill holes along the edge of the colony. The discs were inoculated on 1/4 SDAY in 90.0 mm Petri dishes in a chamber (SPX-420B, Shanghai Nanrong laboratory equipment Co., Ltd., Shanghai, China) for 14 d at 15 °C, 18 °C, 21 °C, 24 °C, and 27 °C. Every sample was processed in triplicate.

#### 2.2.2. Determination of Optimized Nitrogen Sources for the Fungus

According to the complete randomized design, based on Czapek–Dox medium (sodium nitrate 3.0 g/L, potassium dihydrogen phosphate 1.0 g/L, magnesium sulfate heptahydrate 0.5 g/L, potassium chloride 0.5 g/L, ferrous sulfate 0.01 g/L, sucrose 30.0 g/L, agar 20.0 g/L), the sodium nitrate was replaced with different nitrogen sources; the specific amount was shown below: soybean power (9.5 g/L), peptone (4.0 g/L), silkworm pupa powder (7.3 g/L), ammonium nitrate (1.4 g/L), ammonium chloride (1.9 g/L), yeast extract (3.9 g/L). No nitrogen source was added to the control group. Then, the 7.0 mm punch was used to drill holes along the edge of the colony. The discs were inoculated on different nitrogen sources medium. They were then grown in an incubator at 24 °C, out of the light. Their diameters were measured after 14 d and their growth rates were calculated. Every sample was processed in triplicate.

#### 2.2.3. Determination of Optimized Carbon Sources for the Fungus

According to the complete randomized design, based on the Czapek–Dox medium, the sucrose was replaced with different carbon sources; the specific amount was shown below: glucose (31.6 g/L), maltose (30.0 g/L), lactose (30.0 g/L), D-mannitol (31.6 g/L), fructose (31.6 g/L), D-sorbitol (31.6 g/L). Then the 7.0 mm punch was used to drill holes along the edge of the colony. The discs were inoculated on different carbon sources medium. In addition, no carbon source was added to the control group. They were then grown in an incubator at 24 °C, out of the light. Their diameters were measured after 14 d and their growth rates were calculated. Every sample was processed in triplicate.

#### 2.2.4. Determination of Optimized Liquid Medium for the Fungus

The strains were cultured in 1/4 SDAY liquid medium at 145 rpm/min at 24 °C until the concentration of spores in the medium was 1 × 10^7^ spore/mL. Then, the prepared spore suspension was inoculated in five different media with a volume of 150 mL and at a 5% dose (7.5 mL). The medium used in this study was: A (potato 20.0 g/L, glucose 20.0 g/L), B (D-mannose 22.0 g/L, yeast extract 2.0 g/L, KH_2_PO_4_ 1.0 g/L, MgSO_4_ 0.5 g/L), C (glucose 22.0 g/L, yeast extract 2.0 g/L, KH_2_PO_4_ 1.0 g/L, MgSO_4_ 0.5 g/L), D (glucose 20.0 g/L, tussah pupa powder 10.0 g/L, KH_2_PO_4_ 1.5 g/L, MgSO_4_ 1.2 g/L), E (peptone 10.0 g/L, yeast extract 10.0 g/L, glucose 20.0 g/L) [33]. Each medium was carried out in triplicate. Every 72 h, 4 mL culture medium was taken, 3 mL of which was centrifuged at 10,000 rpm for 10 min. Precipitation was collected and dried with absorbent paper, and the weight of precipitation was measured (m_1_). After lyophilization, the weight of fermentation was measured again (m_2_). The water content of metabolism was calculated by the formula: water content (W%) = [1 − (m_1_ − m_2_)/m_1_] × 100. Then we counted the number of spores in the culture medium using a hemocytometer.

### 2.3. Bioassay for Insecticidal Activity

The fungus was cultured at 24 °C on 1/4 SDAY for 14 d to obtain mycelium. After that, an appropriate amount of mycelium was taken with a sterile needle and inoculated in the broth of D in a shaker cultured for 14 d to obtain conidium. The conidium suspension was harvested from the cultured broth by centrifugation (10,000 rpm, 5 min). The centrifuged spores were suspended with sterile water containing 0.5% glycerin and 0.01% Tween-20 into spore suspension, which have been diluted into the concentrations of 1 × 10^3^, 1 × 10^4^, 1 × 10^5^, 1 × 10^6^, 1 × 10^7^ spore/mL and 50 mL for each concentration. The control group was given sterile water only containing 0.5% glycerin and 0.01% Tween-20.

The insects tested in this study were *Henosepilachna vigintioctopunctata*, *Spodoptera exigua*, *Plutella xylostella*, *Spodoptera frugiperda*, *Sitobion avenae*, *Hyalopterus perikonus*, and *Aphis citricola*. The third instar larva of similar size was selected for each species in 18 groups of 15. The insects were fed in an 11.0 × 8.0 × 5.4 (cm, length × width × height) box. The prepared spore suspension was sprayed into the box evenly, and every box received 1 mL. Some food was placed in each box, and insect deaths were counted every three days. 

### 2.4. Statistical Analysis

All the bioassay experiments were performed independently in triplicates. The lethal concentrations (LC_50_, LC_90_) were determined using PoloPlus 2.0 [34]. Our data were analyzed using SPSS, and our presented mean was ±1 standard error of the mean. *p* < 0.05 was considered statistically significant. 

## 3. Results

### 3.1. Isolation and Identification of Strain

The strain QHLA was isolated from the Chinese caterpillar fungus complex (*Cordyceps sinensis*) samples collected in Guoluo, Qinghai Province. The strain QHLA was cultured for 30 d on 1/4 SDAY at 15 °C. The colony diameter was 4.5 cm and the middle of the obverse of the colony was light yellow and the surrounding area was white. The mycelia were dense and villous. The reverse of the colony was yellow and the edge was white (Figure 1a,b). Mycelia were colorless, linear, or spiral, without septum, branching, and 0.5–2.0 μm in diameter. The conidia were elliptic to short clubbed, (2.5–5.0) × (1.0–1.5) μm (Figure 1c–e). The phylogenetic tree was established with *Simplicillium lanosoniveum* CBS704.86 as the outgroup (Figure 2). The strain QHLA and *L. uredinophilum* KUN101466 were clustered in the same branch, and the results of the pairwise sequence (p-)distance analysis were shown in the Appendix A. After morphological and molecular identification, QHLA was *L. uredinophilum*.

### 3.2. Optimum Culture Temperature for the QHLA

The growth rate of the strain QHLA growing at 15–24 °C was increased in a temperature-dependent manner, while significant inhibition was observed when the temperature reached 27 °C. The relationship between mycelium growth rate and the temperature was shown in Figure 3. When the temperature was 15 °C, the obverse side of the colony was white and the reverse side was milky-white. At 18 °C, the obverse side of the colony was white, with some yellow aerial mycelia in the middle, and the reverse side was light yellow. The obverse side of the strain was white, the middle circle was orange, and the reverse side was yellow, when the temperature was 21 °C and 24 °C. The mycelia shriveled and grew into the culture medium; it was brown along with the plate and yellow on the reverse side when the temperature was 27 °C.

### 3.3. Effects of Different Nitrogen Sources on Growth Rate and Sporulation Quantity of the Strain QHLA

QHLA grew normally on eight different kinds of media. The growth rate of the QHLA was yeast extract > silkworm pupa powder > soybean power > peptone > NaNO_3_ > NH_4_NO_3_ > NH4Cl > no nitrogen. The highest mycelial growth rate of QHLA was observed on yeast extract and except for the silkworm pupa powder (there was no significant difference between them), the growth rate was significantly difference from that of other media. The spore production of the QHLA was the highest in the medium with NH4Cl as a nitrogen source, which was significantly different from other media. No spores were produced in a medium without any nitrogen source (Table 1, Figure 4). 

### 3.4. Effects of Different Carbon Sources on Growth Rate and Sporulation Quantity of the Strain QHLA

QHLA grew normally on eight different kinds of media. The growth rate of QHLA was maltose > fructose > D-mannitol > lactose > glucose > sucrose > D-sorbitol > no carbon (Figure 2). The highest mycelial growth rate of QHLA was observed on maltose; but there was no significant difference between maltose, fructose, D-mannitol, lactose, and glucose. The spore production of the QHLA was the highest in the medium with D-sorbitol as a nitrogen source, which was significantly different from other media. No spores were produced in a medium without any carbon source (Table 2, Figure 5).

### 3.5. Optimum Liquid Medium for the QHLA

In the five liquid culture, the QHLA grew mycelia and produced spores, but the yield of mycelia and spores varied with the different media (Figure 6). The mycelia water content of the QHLA was the lowest in medium A, at 57.15 ± 0.54%, and was the highest in medium E. The mycelium production was the highest in medium D, which was 8.78 ± 0.69 mg/mL and the lowest in medium E. The spore production was significantly higher in medium E than in other media, at 3.71 ± 0.52 mg/mL, and the lowest in medium B.

With the increase in incubation time, the mycelial water content; mycelium production; and spore production of the QHLA also changed. The water content of mycelia was the highest at the initial stage of culture, gradually decreasing with the advance of culture time, and it reached its lowest point at the 12th d of culture. After that, the water content of mycelia increased again. Both mycelium production and spore production had a trend of first increasing and then decreasing during the incubation time. Mycelium production reached the highest value on the 9th d, while spore production reached the highest value on the 12th d.

### 3.6. Bioassay

The QHLA had the best biocontrol effect on *S. avenae* and *A. citricola*, with LC_50_ of 6.32 × 10^3^ spore/mL and 6.35 × 10^3^ spore/mL, respectively. However, as the concentration of spores increased, the QHLA had different control effects on the two aphids. LC_90_ of the QHLA on *A. citricola* was 2.11 × 10^7^ spore/mL and 2.36 × 10^8^ spore/mL on *S. avenae* (Table 3).

## 4. Discussion

The Chinese caterpillar fungus (*C. sinensis*), a well-known entomopathogenic fungus in traditional Chinese medicine, is a complex formed by *Ophiocordycepes sinensis* parasitic larvae of the ghost moth (*Hepialus armoricanus* Obertheir, Lepidoptera) [35]. Previous experiments have also isolated a variety of other fungi, such as *Isaria feline*, *Samsoniella hepialid*, *I. farinose*, *I. fumosorosea* [36,37]. In addition, the fungi of the *Lecanicillium* sp. come from a wide range of sources and are widely distributed. For example, *L. cauligalbarum* emerges from the corpse of a stemborer (Lepidoptera) [38], and *L. primulinum* was isolated from soil in Okinawa’s main island and the Bonin Islands [39]. *L. subprimulium* originates from decaying wood in Baoshan City, Yunnan Province, China [1]. 

In this study, the strain QHLA was isolated from *C. sinensis*. After phylogenetic analysis of a combined data set comprising *ITS*, nr*SSU*, nr*LSU*, *RPB1*, and *RPB2,* sequence data supported that QHLA was *L. uredinophilum*. At the same time, the morphological result of QHLA was consistent with the results described by Zare and Gams for the *Lecanicillium* sp. [40]. The *L. uredinophilum* was first reported in Korea and was isolated from a rust fungus [16]. In China, *L. uredinophilum* was first obtained from an infected insect collected in Yunnan Province [17].

Environmental factors, especially temperature and humidity, affect the growth rate of fungi [41,42]. The optimum culture temperature for *Lecanicillium* sp. is 24 °C, and for the strain, QHLA is between 21 °C to 24 °C [43]. In addition to temperature and humidity, nutrition, mineral elements, vitamins, and other cultural conditions also have a great influence on the growth rate and sporulation of fungi, and the *Lecanicillium* sp. fungi are cultured in a variety of media. For example, *L. lecanii* CA-A-G is favorable for sporulation at 29 °C, a pH of 4, and under full light. However, when the temperature is 23 °C, the pH drops to 3, and photology is 12L:12D, the environment is more conducive to the accumulation of mycelia and other biomass of the strain [44]. *L. lecanii* SN21 is cultured in a variety of media, and a medium with a high C/N ratio is favorable for producing spores [45]. 

*Lecanicillium* sp. are entomopathogenic fungi that invade host insects by secreting cell wall-degrading enzymes, such as chitinase and lipase, and then its host insects are effectively killed [46,47]. *L. attenuatum* was reported to be effective in controlling cypress aphids (*Cinara cupressi*) with LC_50_ 3 × 10^5^ spore/mL [48], and *L. attenuatum* also has a control effect on cotton aphid (*A. gossypii*) [49]. The LT_50_ for *L. attenuatum* and *L. longisporum* to *M. persicae*, *Macrosiphum euphorbiae*, and *Aulacorthum solani* are between 2 d to 4 d, indicating a potential for biological control [50]. 

*L. muscarium* works with *A. nigripes* to enhance virulence against Potato Aphid [51], and strains of *L. muscarium* are also used for the control of *B. tabaci*, *T. vaporariorum*, *A. craccivora*, and *Pristiphora abietina*. Moreover, the activity of *L. muscarium* to *Aphis craccivora* is higher than that of *M. pingshaense* and *B. Bassiana* [6,52,53]. The V-3 strain of *L. lecanii* has a high control effect on whitefly (*B. tabaci*), with a maximum mortality rate of 90.6%. In addition, the filtrate of the V-3 strain also controls *B. tabaci* [11], and the temperature at which *L. lecanii* is most active against *B. tabaci* is 24 °C [54].

*L. lecanii* SN21 is also used to control *Frankliniella occidentalis* (Thysanoptera: Thripidae) [39]. However, different *L. lecanii* and *L. muscarium* have varied virulence to *T. castaneum*, and LC_50_, which ranged from 2.83 × 10^5^ to 6.133 × 10^22^ spore/mL [55]. According to current research results, *Lecanicillium* sp. has prospective application potential in biological control, especially for Hemiptera control. In addition, the virulence of *Lecanicillium* sp. against various biological stages of *T. vaporariorum* was enhanced after protoplasmic fusion [56]. 

There are many species of entomogenous fungi that are used for aphid biological control, and our study focused on the insecticidal range of *L. uredinophilum*. We found that the strain of *L. uredinophilum* QHLA has the best control effect on aphids, especially on *S. avenae* and *A. citricola*. The LT_50_ of *Pandora neoaphidis* and *Entomophthora planchoniana* for *S. avenae* was about 5 d, which merits further study of the QHLA [57].

## 5. Conclusions

A new strain QHLA of *L. uredinophilumwas* was isolated from the Chinese caterpillar fungus. The QHLA was highly pathogenic against *S. avenae* and *A. citricola* with LC_50_ of 6.32 × 10^3^ spore/mL and 6.35 × 10^3^ spore/mL, respectively. For optimized culture conditions, the best sporulation was produced with NH_4_Cl as the nitrogen source and D-sorbitol as the carbon source, while the best mycelial growth rate was obtained with the yeast extract as the nitrogen source and maltose as the carbon source. For the liquid culture of the strain QHLA, peptone; yeast extract; and glucose were necessary for the best spore production with a fermentation period of 12 days, while glucose, tussah pupa powder, KH_2_PO_4_, and MgSO_4_ were necessary for the highest mycelia production with a fermentation period of 8 days. 

## Figures and Tables

**Figure 1 microorganisms-10-01832-f001:**
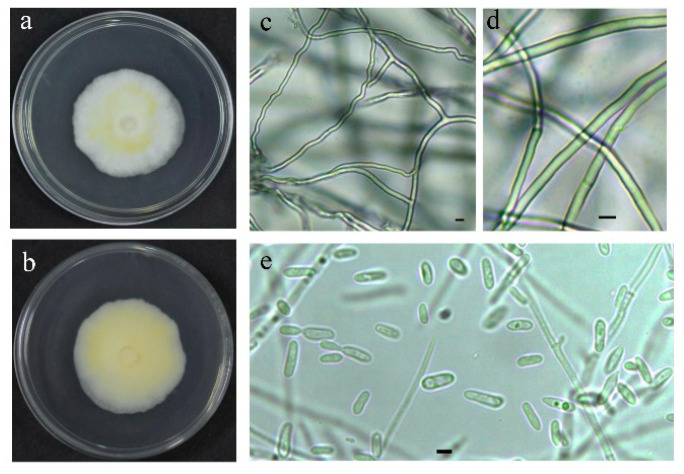
The characteristic features of the QHLA in a petri dish: (**a**,**b**). The microscopic features of the QHLA: (**c**–**e**). Bars = 2.5 μm.

**Figure 2 microorganisms-10-01832-f002:**
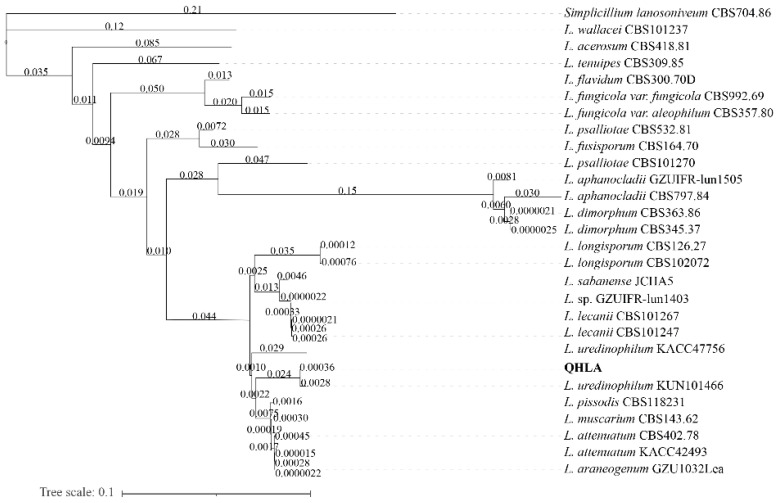
Phylogenetic tree based on *ITS* nr*SSU* nr*LSU RPB1* and *RPB2* sequences by using PhyloSuite v1.2.1.

**Figure 3 microorganisms-10-01832-f003:**
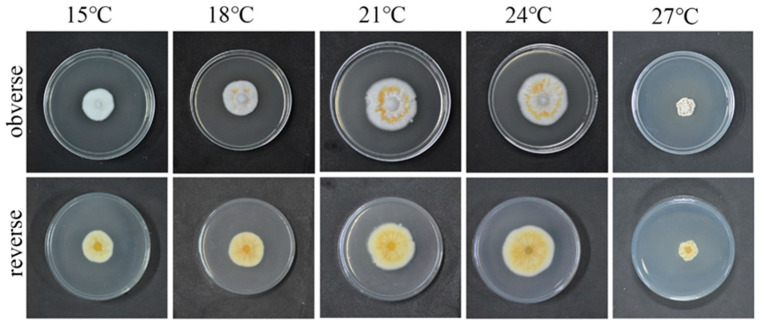
QHLA growth at different temperatures.

**Figure 4 microorganisms-10-01832-f004:**
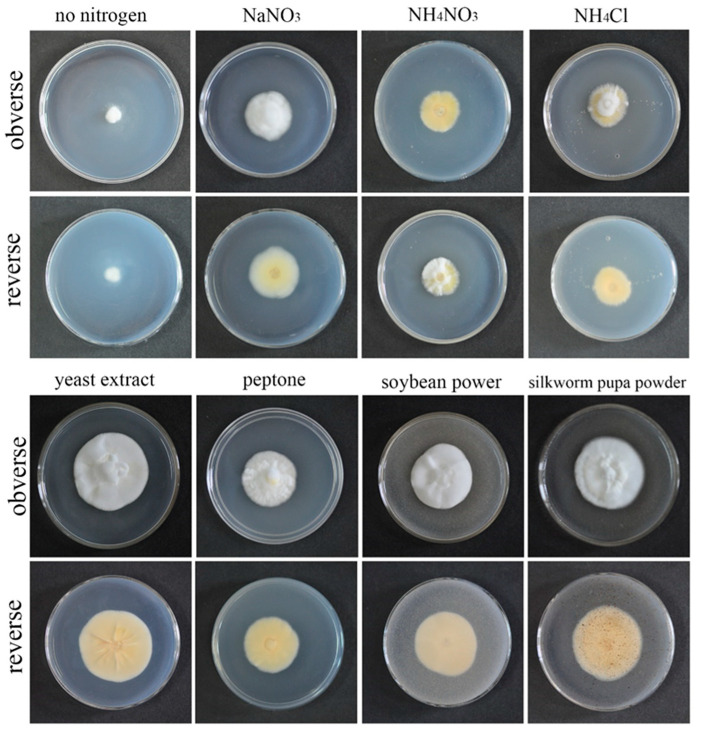
Growth of the QHLA on different media with different nitrogen sources.

**Figure 5 microorganisms-10-01832-f005:**
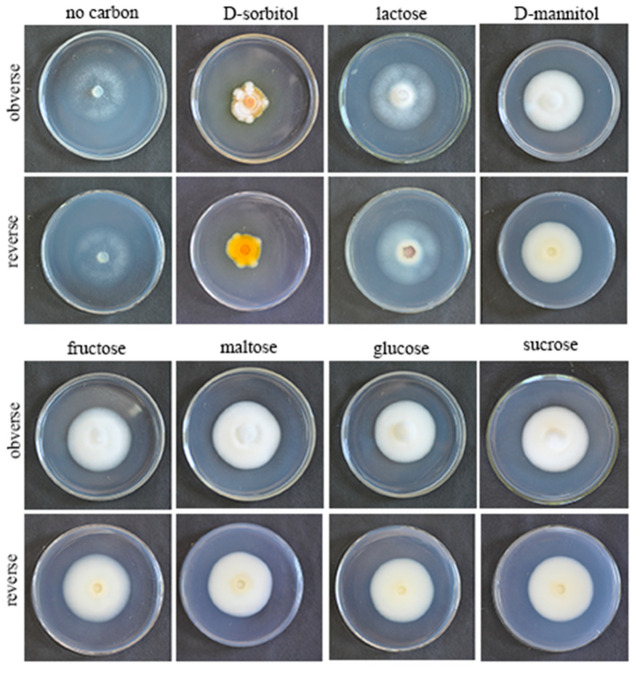
Growth of the QHLA on different media with different carbon sources.

**Figure 6 microorganisms-10-01832-f006:**
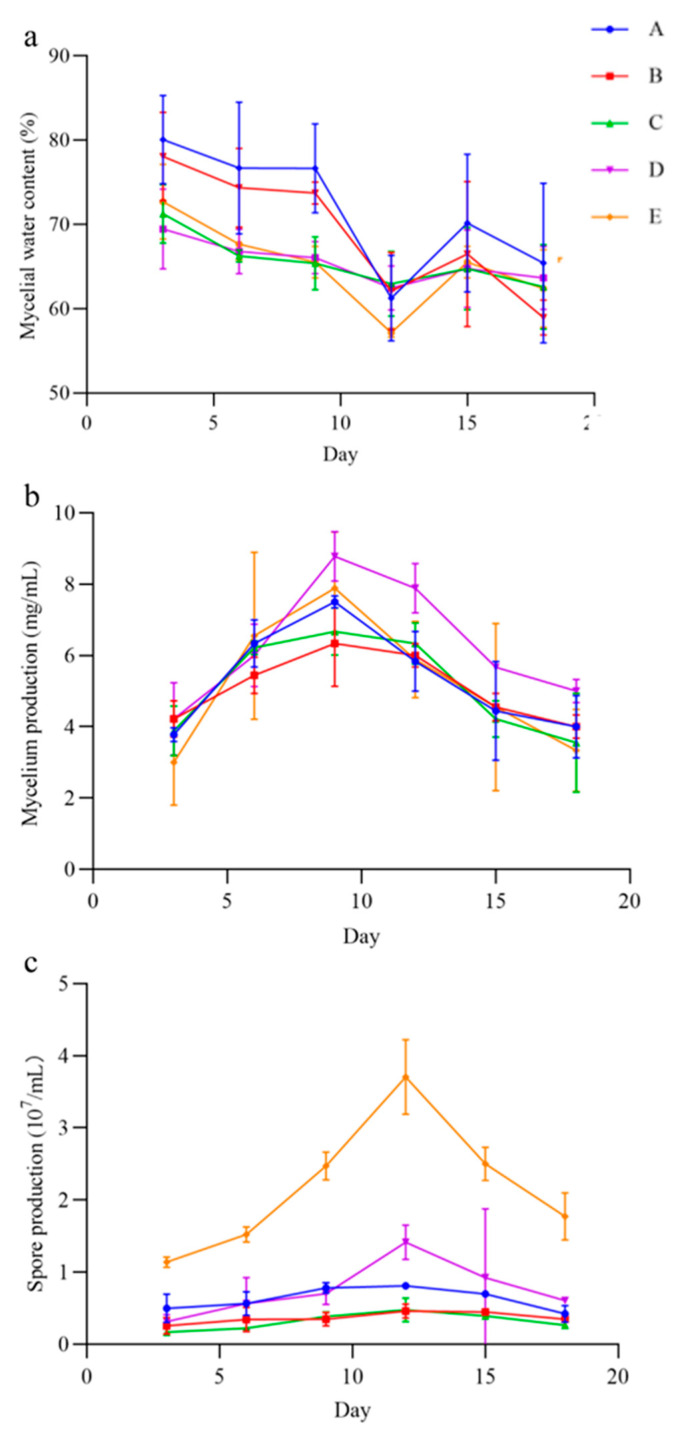
The mycelia water content, mycelium production, and spore production in different media of the QHLA. A (potato, glucose), B (D-mannose, yeast extract, KH_2_PO_4_, MgSO_4_), C (glucose, yeast extract, KH_2_PO_4_, MgSO_4_), D (glucose, tussah pupa powder, KH_2_PO_4_, MgSO_4_), E (peptone, yeast extract, glucose). Note: (**a**): mycelial water content of the QHLA; (**b**): mycelium production of the QHLA; (**c**): spore production of the QHLA.

**Table 1 microorganisms-10-01832-t001:** Colony growth rate, spore production, and colony morphology of the QHLA on different nitrogen sources.

Nitrogen Source	Colony Growth Rate (mm/d)	Spore Production (spore/mL)	Colonial Morphology	Mycelial Growth Vigor
Yeast extract	4.63 ± 0.03 ^a^	1.15 × 10^6 d^	The obverse side of the colony was white, and the reverse side was yellow with a white edge.	+++
Silkworm pupa powder	4.61 ± 0.19 ^a^	1.85 × 10^6 cd^	The obverse side of the colony was white, and the reverse side was pale yellow with a white edge.	+++
Soybean power	4.03 ± 0.20 ^b^	2.40 × 10^6 bc^	The obverse side of the colony was white, and the reverse side was pale yellow with a white edge.	+++
Peptone	3.33 ± 0.56 ^c^	2.02 × 10^6 cd^	The obverse side of the colony was white, and the reverse side was pale yellow with a white edge.	++
NaNO_3_	3.00 ± 0.06 ^d^	1.40 × 10^6 cd^	The obverse side of the colony was white, and the reverse side was pale yellow with a white edge.	++
NH_4_NO_3_	2.24 ± 0.06 ^e^	2.74 × 10^6 b^	Both sides of the colony were golden yellow with white aerial mycelia.	+
NH_4_Cl	2.22 ± 0.20 ^e^	4.08 × 10^6 a^	The obverse of the colony was white in the middle with white aerial mycelia and yellow around it. The reverse side was yellow.	+
−	0.52 ± 0.18 ^f^	−	The colony was no obvious mycelium on both sides	+

Note: ‘+’ indicates that mycelia are spares and slim, ‘++’ indicates that mycelia are dense and strong, ‘+++’ indicates that mycelia are thick. Different lowercase letters are expressed as the extremely significant level of 5%. −: no nitrogen.

**Table 2 microorganisms-10-01832-t002:** Colony growth rate, spore production, and colony morphology of QHLA on different carbon sources.

Carbon Source	Colony Growth Rate (mm/d)	Spore Production (spore/mL)	Colonial Morphology	Mycelial Growth Vigor
Maltose	2.65 ± 0.08 ^a^	7.17 × 10^5 b^	The obverse of the colony was white. The middle of the reverse was light yellow and the edge was white.	+++
Fructose	2.64 ± 0.06 ^a^	7.17 × 10^5 b^	The obverse of the colony was white. The middle of the reverse was milky colored with white edges.	++
D-mannitol	2.56 ± 0.25 ^ab^	5.33 × 10^5 b^	The middle of the obverse of the colony was yellow and the edge was white. The middle of the reverse was white, and the edge was pale yellow with white aerial mycelia.	++
Lactose	2.51 ± 0.11 ^ab^	9.33 × 10^5 b^	The obverse of the colony was white, and there were dense aerial mycelia around the disk. The reverse was brown in the middle and white around the edges.	+
Glucose	2.45 ± 0.24 ^ab^	7.59 × 10^5 b^	The obverse of the colony was white, and the middle of the reverse was milky colored with white edges.	+++
Sucrose	2.35 ± 0.11 ^b^	4.83 × 10^5 b^	The obverse of the colony was white, and the middle of the reverse was light yellow with white edges.	+++
D-sorbitol	2.01 ± 0.10 ^c^	1.63 × 10^6 a^	The obverse of the colony was orange, with white aerial mycelia above. The reverse was orange.	+
−	0.31 ± 0.13 ^d^	−	Colonies were white on both sides.	+

Note: ‘+’ indicates that mycelia are spares and slim, ‘++’ indicates that mycelia are dense and strong, ‘+++’ indicates that mycelia are thick. Different lowercase letters are expressed as the extremely significant level of 5%. −: no carbon.

**Table 3 microorganisms-10-01832-t003:** The LC_50_ and LC_90_ of the QHLA.

Pests	LC_50_ (spore/mL)	95% Confidence Interval	LC_90_ (spore/mL)	95% Confidence Interval
*Henosepilachna vigintioctopunctata*	4.83 × 10^5^	7.811–3169.280	7.12 × 10^10^	3.51–∞
*Spodoptera exigua*	1.32 × 10^5^	2.001–970.003	1.39 × 10^9^	1.492–∞
*Plutella xylostella*	4.83 × 10^5^	7.811–3169.280	7.12 × 10^10^	3.51–∞
*Spodoptera frugiperda*	4.83 × 10^5^	7.811–3169.280	7.12 × 10^10^	3.51–∞
*Sitobion avenae*	6.30 × 10^3^	0.215–1.713	2.36 × 10^8^	0.187–247.71
*Hyalopterus perikonus*	1.49 × 10^4^	0.826–2.260	1.88 × 10^9^	12.215–34.82
*Aphis citricola*	6.40 × 10^3^	0.042–1.923	2.11 × 10^7^	0.0340–28.24

## Data Availability

The sequences of QHLA (Accession numbers are listed in Appendix A) are available in the Genack datable.

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
