# Peer review of "A New Strain of Lecanicillium uredinophilum Isolated from Tibetan Plateau and Its Insecticidal Activity"

_microorganisms, 2022, doi:10.3390/microorganisms10091832_

Round 1

Reviewer 1 Report

The authors report on the isolation of a fungal entomopathogen, its morphological and molecular identification as Lecanicillium uredinophilum as well as growth and virulence characteristics.

Concerning the text in general:

There is alongside the manuscript a problem with blanks that obviously got lost: "L. uredinophilumwas", "significantly[11]", ...

Superscripts in scientific notation should be presented as such, not as, e.g., "concentrations of 103, 104, 105, 106, 107 spore/mL"

Unfinished sentence in line 47.

Genus and species designations should be in italics.

Abbreviated species designations should carry the first letter only of the genus name plus the species epithet: "M. pingshaense" instead of "Met. pingshaense" etc.

Introduction section:

There could be a piece of information referring to L. uredinophilum. This information follows only in the discussion section.

The presentation of Methods - Results - Discussion is generally cosistently structured.

Presentation and discussion of microbiological and bioassay data is fine. Concerning the phylogeny part see below.

Table 3: It might be easier to grasp the meaning of data if LC50 and LC90 values were indicated in spores/mL, with the complete value written in scientific notation in the table, as done in "spore production" columns in Tables 1 and 2. Moreover, insect species names could be given in full.

References:

Generally OK, works are duly cited. Reference 51 gives first and not family names of the authors, this should be changed.

Phylogeny part of the study:

It is hard to tell from the presentation of the phylogeny part of the manuscript if identification of isolate QHLA as L. uredinophilum is well founded or not.

Figure 2 shows next to no resolution of relevant clades. This is probably mainly due to the choice of a very distantly related outgroup. Choosing are closer outgroup taxon (e.g. from the genus Beauveria) might be helpful.

Moreover, the lower clade of the tree does not appear to be relevant and could be entirely removed.

Neither the methods section nor the legend to figure 2 explain how the tree has been generated from the 5 marker sequences aligned. Is it a consensus tree from 5 individual marker trees or a tree generated from concatenated marker sequences?

Overall, the figure  legend could be more informative, indicating what numbers on branches or the size bar designate.

Bootstrap support of the relevant clade consisting of L. uredinophilum KACC and isolate QHLA is indicated as 78% (?). This value is very low, a minimum requirement for conclusions concerning species identification would be >90%, better >95%.

Moreover, authors indicate that "similarity was 84%". Firstly this might be a very low value, too, and secondly the methods section does not give any indication that sequence distance analysis has been performed (number of differences, p-distances, Jukes-Kantor model, ...). 

The discussion section states that L. uredinophilum has previously been described twice. Sequences from the second description (reference 36) are available in Genbank (strain KUN 101466 etc.), but have not been used for comparison. This should imperatively be done.

In conclusion, presentation and content of the phylogenetic analysis should be considerably revised. The respective methods section should give more detailed information on the bioinformatic analses performed. To come to a sound identification of the new isolate, both phylogenetic reconstruction with bootstrapping and pairwise sequence (p-)distance analysis at least for markers ITS, RPB1 and RPB2 comprising QHLA and most closely related Lecanicillium species might appropriately be performed. All this can be easily done in the MEGA software tool already employed by the authors.

Reviewer 2 Report

The manuscript entitled „A new strain of Lecanicillium uredinophilum isolated from Tibetan Plateau and its insecticidal activity” describes a novel strain of Lecanicillium sp., which according to the performed molecular analysis could be assigned as L. uredinophilum. The Authors characterized the optimal condition of the strain growth and tested its insecticidal activity towards seven different species. Their results suggest that the strain could be considered as an interesting candidate for new microbial pesticide. However, further studies are needed.

The subject of the manuscript is interesting and important, although the quality of the manuscript and results presentation require major improvement from the editorial point of view. The Authors should remember that Latin names should be always in italic and that there are some abbreviation rules. Some names were abbreviated even during first use (i.e. Beauveria, Metarhizium), some of them were abbreviated in strange way… Also the chemical formulas (subscripts), numbers (superscripts) and units should be carefully checked before submitting manuscript to any journal. The Authors should pay more attention to numbering the Figures, referring Figures and Tables… These shortcomings make reading the manuscript difficult and unpleasant. The publication looks sloppily written, which has a negative impact on the perception of manuscript. It is not a big effort to omit these errors at the reading stage before submission.

Some detailed comments below:

l29. thean? the or an?

l30 delete it

l45 explain JA and SA in the text

l60. maybe some coordinates to be more specific?

Paragraphs 2.2.2 and 2.2.3 – it would better to rephrase these parts, now they are not clear (in contrary to 2.2.5 which is well-written). Maybe change into: the optimal sources (C/N) were determined using Czapek-Dox medium with different compounds (list them) as C/N source. Now there is nothing about inoculation…

Figure 1 – usually the figures and panels are ordered according to their appearance in text (d-e should become a-b).

Figure 2 – gaps were removed? what are the lengths of each genes fragments used to build a tree?

l192 I believe Figure 1 should be Table 1

l194 significantly what?

Table 1 – use the same decimal number within one parameter

l210 I believe Figure 2 should be Table 2

There are no references for Figure 4 and 5 in the main text.

l226, l228 and l229 there is no need to show so many decimal numbers, two will be enough

Figure 6 add to the figure’s description in l240 after media “(A-E)”

l280 I believe chitinase instead of chitin

Round 2

Reviewer 1 Report

The authors have reacted positively to rthe previous suggestions and comments.

Especially introduction of further L. uredinophilum reference sequences and p-distance analysis appear to provide a more solid basis for the main phylogenetic conclusion than in the earlier version of the manuscript.

However, the L. uredinophilum reference sequences added to the anaylsis (strain KUN 101466) are not listed in Table S2. As relationship to this strain is the crucial argument derived from phylogenetic reconstruction, this should be corrected.

Concerning the clade structure resolution and scientific infiormation presented, Figure 2 has been considerably improved with respect to the earlier version. However, numbers on branches and strain designations are hard to read, font size might be improved. Just a lay-out problem, not a scientific one.

Citation indications on the first page's left margin might ne erroneous.

Author Response

Dear Reviewer,

Thank you very much for your careful and helpful comments!

Comments and Suggestions for Authors

1.The L. uredinophilum reference sequences added to the anaylsis (strain KUN 101466) are not listed in Table S2. As relationship to this strain is the crucial argument derived from phylogenetic reconstruction, this should be corrected.

Response: The reference sequences of KUN 101466 have been added in Table S2.

  1. Numbers on branches and strain designations are hard to read, font size might be improved.

Response: We corrected as your suggestion. All font size was enlarged.

Sincerely,

Dr. Dun Wang, the corresponding author